# Size Effect on Mechanical Properties and Deformation Behavior of Pure Copper Wires Considering Free Surface Grains

**DOI:** 10.3390/ma13204563

**Published:** 2020-10-14

**Authors:** Yu Hou, Xujun Mi, Haofeng Xie, Wenjing Zhang, Guojie Huang, Lijun Peng, Xue Feng, Zhen Yang

**Affiliations:** 1State Key Laboratory of Nonferrous Metals and Processes, GRIMAT Group Co., Ltd., Beijing 100088, China; beifeitianxia@126.com (Y.H.); zhangwenjing@grinm.com (W.Z.); huangguojie@grinm.com (G.H.); penglijun198677@163.com (L.P.); yzcopper@163.com (Z.Y.); 2GRIMAT Engineering Institute Co., Ltd., Beijing 101407, China; fengxue@grinm.com; 3General Research Institute for Nonferrous Metals, Beijing 100088, China

**Keywords:** size effect, pure copper wires, mechanical properties, deformation behavior, surface model, free surface grain

## Abstract

The size (grain size and specimen size) effect makes traditional macroscopic forming technology unsuitable for a microscopic forming process. In order to investigate the size effect on mechanical properties and deformation behavior, pure copper wires (diameters range from 50 μm to 500 μm) were annealed at different temperatures to obtain different grain sizes. The results show that a decrease in wire diameter leads to a reduction in tensile strength, and this change is pronounced for large grains. The elongation of the material is in linear correlation to size factor D/d (diameter/grain size), i.e., at the same wire diameter, more grains in the section bring better plasticity. This phenomenon is in relationship with the ratio of free surface grains. A surface model combined with the theory of single crystal and polycrystal is established, based on the relationship between specimen/grain size and tensile property. The simulated results show that the flow stress in micro-scale is in the middle of the single crystal model (lower critical value) and the polycrystalline model (upper critical value). Moreover, the simulation results of the hybrid model calculations presented in this paper are in good agreement with the experimental results.

## 1. Introduction

With the rapid development of micro-electro-mechanical systems (MEMS) and their gradual application, the demand for micro-components is increasing. Plastic micro-forming technology has a wide application prospect in the fields of micro-mechatronics, bio-medicine and micro-energy due to its advantages of simple processing, high efficiency, low cost, excellent mechanical properties, and good repeatability. In plastic micro-forming technology, the size of the molded parts is less than 1 mm [1,2,3,4], with the crystal model ranged from single crystal to polycrystal. As a result of the size effect, the forming quality and basic properties of the miniature parts are changed [5,6,7]. This change differs from the class of materials including crystalline and amorphous metals. Some literature mentions that materials have distinct critical size points, i.e., changes in mechanical properties at grain sizes less than 500 nm, specimen sizes less than 100 μm, or sample size to grain size ratios less than 1, and that these parameters constrain each other, making it difficult to specify the exact size effect transition point [8,9].

Researchers conducted a lot of investigations in the early stage. Geiger et al. [10,11] conducted some experimental studies on the size effect of flow stress in microscopic coarse deformation, and proposed a surface model to explain the phenomenon. Through a series of stretching and flanging experiments of geometrically similar parts, the experiment of Kals, Li and Engel [7,12,13] displays this behavior for CuNi18Zn20 and CuZn37, an average grain size (d) of approximately 40 μm for different values of the sample size. They proposed a “surface model” in which, when the microstructure of the specimen remains unchanged, the ratio of surface-layer grains and the internal-layer grains increase with the specimen size decreasing [5], dislocation cannot be accumulated in the specimen surface, there is reduced material work hardening ability and this thus results in the decrease of the flow stress of material. In recent studies, it has been observed that the yield strength and tensile strength decrease as T/D (thickness/average grain diameter) ratio [8,14] decreases, and in Gau’s study, this trend continued until T/D fell to 1 [15]. In general, when the specimen size is changed by plastic deformation, the internal grain size also changes. The effect of grain refinement on strengthening is interwoven with the specimen size. Simple experimental comparison cannot reveal the size effects [5,16].

In this work, experimental investigations are performed on different diameters copper wires with different grains by tensile tests, in order to study the influence of specimen size (D), grain size (d) and the ratio (D/d) on the mechanical properties and flow behavior, respectively. The size effects on the flow stress curves are clearly demonstrated and can be used to validate a new model of the constitutive behavior.

## 2. Experimental

Copper fine wires were drawn from a laboratory ingot, using commercial cathode copper (Jiangxi Copper Groups, Guixi City, Jiangxi, China) as raw material, the measure of the copper ingot was φ30 mm × 180 mm, weighting 1.85 kg. The copper ingot with a purity of 99.93 wt.% (impurity content: 0.03 wt.% Ag, 0.02 wt.% O, 0.01 wt.% S and other impurities which are less than 0.01 wt.%) was drawn in varied dimensions (50, 70, 100, 200, 300 and 500 μm), and they were annealed in a vacuum for 1 h at temperatures ranging from 400 °C to 650 °C to obtain different grain sizes. The specimens for microstructural observation were vertically inlayed with a mounting press machine (Yuzhou ZXQ-1, Laizhou, Shandong, China). The inlayed specimens were sanded and polished to make its cross-section exposed, and then etched in the reagent (10 wt.% FeNO_3_ + 90 wt.% CH_3_COOH). Microstructural observation was performed using an optical microscope (OM: Zeiss Axiovert 200 MAT, ZEISS Group, Oberkochen, Germany), and the grain sizes were measured with the microstructural information, using the standard method of GB/T 6394-2017.The microstructures of the annealed specimens are shown in Figure 1 and the grain sizes obtained are shown in Table 1. 

The gauge length of the tensile specimen was 100 mm referring to the ASTM standard E8/E8M-13a. The tensile tests were performed on an Instron testing machine at a low strain rate of five mm/min. Abrasive paper was placed between the specimen surface and the fixture to increase the friction, avoiding fractures at the clamping position.

## 3. Results

The tensile strengths of specimens with different grain and specimen sizes are show in Figure 2. As wire diameter decreases, the tensile strength of the specimen with similar grain size decreases obviously, and this downward trend is more pronounced with large grains. According to the traditional fine-grain intensification theory, the tensile strength of the sample decreases as the grain size increases. The change of grain size has little effect on the tensile strength of the 500 μm wire diameter specimen, however, this change becomes apparent on smaller wire diameters.

Figure 3 shows the effect of grain size and diameter on flow stress separately. Figure 3a represents the flow stress curves of 200 μm wire diameter specimen with different grain sizes, a negative correlation between yield stress and grain size is observed, which is consistent with the experimental results of previous studies [4,5,17]. With the same diameter, specimens with smaller grain size has higher flow stress, which is the result of fine-grain strengthening. In the elastic stage, flow curves with different grain sizes go up at the same rate, indicating that the change of grain size does not affect the elastic modulus. Figure 3b shows the flow stress curves of specimens with the same grain size (5–6.3 μm), and it indicates that smaller wire diameter can increase the flow stress in the elastic stage, by contrast with the effect of grain size, and the decline of wire diameter increases the velocity of flow stress.

Since the specimen diameter and grain size jointly affect the flow stress, a parameter of the specimen size factor (T/d) is introduced to describe the size variation of sheet specimen, that is, the ratio of sheet thickness to grain size. In wire specimens, a similar size factor can be defined as (D/d), this parameter is a combination of macro and micro factors that can be used to characterize changes in materials.

Figure 4 shows that the smaller the grain size, the greater the elongation at break for a given specimen diameter, and there is a linear correlation between the elongation and the reciprocal of the size parameter (D/d), and the slopes are shown in the figure. This is because the greater the number of grains involved in the deformation of the cross-section, the better the coordinated deformation ability between the grains, delaying the occurrence of fracture [18]. This slope becomes progressively smaller with the specimen size increasing, implying that the dependence of the elongation of the material on D/d gradually decreases [19].

## 4. Discussion

### 4.1. Classification of the Size Effects 

A flow stress curve is the most commonly used method to describe deformation behavior, it can illustrate the forming force, the load on the specimens and the local flow behavior [3,20]. As miniaturization increases, there are two kinds of size effect mechanisms that can affect the behavior of materials [11,21]. One is the specimen size effect and the other is the grain size effect. Figure 5 is the general view of these two kinds of size effect. As shown in Figure 5a, the grain size effects occur with the grain size increasing from d_1_ to d_2_ as the dimension is kept constant. In Figure 5b, as the wire diameter decreases from D1 to D2, the effect of specimen size also appears. The grain size effect has been studied extensively, with the Hall–Petch equation being commonly used. However, specimen size effects are currently the subject of research in micro molding processes.

The decreasing flow stress with the increasing miniaturization can be explained by the so-called surface model (Figure 5). The grains located in the free surface layer are less restricted than the grains inside. Dislocations move through the grains and pile up at grain boundaries but not at the free surface during the deformation process [22,23] and, therefore, this leads to less hardening and lower resistance against deformation of surface grains, and thus, a decrease in the tensile strength. With the diameter decreases, the content of surface grains increases, and the flow stress decreases. The flow stress can be expressed as:(1)$$\sigma =\frac{N\text{\_}s\sigma_{s}+N_{i}\sigma_{i}}{N}\left(N=N_{s}+N_{i}\right)$$

Here, σ is the flow stress of the material; N_s_ and N_i_ are the numbers of grains in the surface and inner layers, respectively; n is the total number of grains; σ_s_ is the flow stress of the surface layer grains, and σ_i_ is the flow stress of the inner layer grains.

Figure 6 shows the grain classification of the cross section of the material, the grain size of which is maintained. It shows that as the specimen size decreases, the deformation behavior shifts from a macroscopic-scale polycrystalline model to a microscopic-scale monocrystalline model. Therefore, the flow stress at the smaller scale of the specimen can be expressed in the following way [24,25]:
(2)$$\sigma_{\mathrm{sig}}\leq \sigma_{\mathrm{micro}}\leq \sigma_{\mathrm{poly}}$$
where σ_sig_ is the flow stress of a single crystal. σ_poly_ is the flow stress of a polycrystal and σ_micro/meso_ is the flow stress for material in micro/meso-scale. For micro-scale deformation modeling, the single-crystal model is the lower limit and the polycrystal is the upper limit.

### 4.2. Simulation of Flow Stress During Micro-Forming

#### 4.2.1. In Single Crystal Model

According to crystal plastic theory and Schmid [6,17,26,27], the critical shear resolved stress of the single crystal can be expressed as:
(3)$$\tau_{R}=\left(\cos\phi \cos\lambda \right)\sigma =\beta \sigma \;(0<\beta \leq \frac{1}{2})$$
in which β is Schmid factor related to the grain orientation; φ is the angle between the normal stress σ and the normal direction of the slip plane; and λ is the angle between the slip direction and the normal stress. Thus, the single crystal model can be described as:(4)$$\sigma_{\mathrm{sig}}\left(\varepsilon \right)=m\tau_{R}\left(\varepsilon \right)$$
in which m is the orientation factor (m > 2).

#### 4.2.2. In Polycrystal Model

The sizing effect of single-crystal yield stress is similar to that of polycrystalline materials, but their reinforcement mechanisms may be completely different, shifting from dislocation source control to dislocation motion control as the size of the sample increases, and the response of single crystals and polycrystals to this control is quite different [28,29]. A smaller single crystal size has a larger free surface ratio, which tends to cause dislocations in order to escape from the sample surface more easily [30]. Therefore, in a smaller sample, additional stress is required to activate the dislocations emitted from a smaller source of dislocations to accommodate the single crystal plastic deformation, which requires more dislocations to slip. In large-sized samples, this additional stress has little effect. 

A large number of grain boundaries exist in polycrystals, which contrast with the free surface in single crystals. These grain boundaries act as barriers that may impede the approach of dislocations, thus creating a series of dislocations around the grain boundaries. Therefore, the reinforcement mechanism of misplaced stacking is effective in polycrystal.

The Hall–Petch equation is one of the most extensive empirical theories between yield stress and grain size [13,26,31], which was further extended by Armstrong [2] to include the flow stress region as follows:
(5)$$\sigma \left(\varepsilon \right)=\sigma_{0}+\frac{k\left(\varepsilon \right)}{\sqrt{d}}$$
in this equation, σ_0_(ε) is known as the friction stress required to move individual dislocations in micro-yielded slip band pile-ups confined to isolated grains, whereas k(ε) is the locally intensified stress needed to propagate general yield across the polycrystal grain boundaries [32]. d is the grain size.

The value of σ_0_(ε) obtained from Hall–Petch analysis is also related with the critical resolved shear stress τ_R_(ε) of a single crystal as [33,34]:(6)$$\sigma_{0}\left(\varepsilon \right)=M\tau_{R}\left(\varepsilon \right)$$

Here, M is the directional coefficient associated with slippage on the deformation system. In the Taylor model [30,35], an upper bound model requires at least five active slip systems. while the lower bound model (the Schmid model), requires only one active slip system [1]. For FCC crystals, M is equal to 3.06 and 2.23 for the Taylor and Schmid modellings, respectively [14,36]. Thus, the polycrystal flow stress can be expressed as follows [26,37,38]:(7)$$\sigma_{\mathrm{poly}}\left(\varepsilon \right)=M\tau_{R}\left(\varepsilon \right)+\frac{k\left(\varepsilon \right)}{\sqrt{d}}$$

#### 4.2.3. Material Behavior Model in Micro-Scale

In wire specimens, the cross-section can be separated into two parts, surface grains in the outer layer and inner grains in the interior part. Size factor (η) represents the proportion of surface grains in the total number of grains.

The ratio of the grain at the surface layer to the total grain in the specimen can be defined as:(8)$$\eta =\{\begin{array}{c} \frac{N_{s}}{N}=\frac{4\left(\frac{D}{d}-1\right)}{{\left(\frac{D}{d}\right)}^{2}}=\frac{4d\left(D-d\right)}{D^{2}}\;\;\;\;D\neq d\\ 1\;\;\;\;\;\;\;\;\;\;\;\;\;\;\;\;\;\;\;\;\;\;\;\;\;\;\;\;\;\;\;\;\;\;\;\;\;\;\;\;\;\;D=d \end{array}$$

Thus, a size factor η is defined, and size factor η is correlated with D/d, but allows for a more intuitive characterization of the proportion of surface grains. 

In this article, based on the above assumptions of the surface model, we propose a hybrid material model. According to the assumptions of the surface model, the whole material is cross-sectionally divided into two different parts: the surface grains and the inner grains. Thus, the flow stress of the material is contributed by two kinds of stress, one from the inner grain and one from the surface grain. Therefore, with the reduction of sample size, the flow stress drops, so the size effects appear. On the microscopic scale, the ratio of these two different grains changes significantly with the manufacturing scale (specimen size), that is to say, whether it is the increase in the ratio caused by the decrease of the sample size or the grain size, the increase in the proportion brought about by the increase will lead to the size effect. From these studies, the hybrid material model consists of two listed components:(9)$$\{\begin{array}{c} \sigma_{s}\left(\varepsilon \right)=m\tau_{R}\left(\varepsilon \right)\\ \sigma_{\mathrm{poly}}\left(\varepsilon \right)=M\tau_{R}\left(\varepsilon \right)+\frac{k\left(\varepsilon \right)}{\sqrt{d}} \end{array}$$
Taken together, the flow stress of the micro-forming process can be expressed in this form:(10)$$\sigma \left(\varepsilon \right)=\frac{N_{s}m\tau_{R}\left(\varepsilon \right)+N_{i}(M\tau_{R}\left(\varepsilon \right)+\frac{k\left(\varepsilon \right)}{\sqrt{d}})}{N}$$

Considering N_s_ = ηN, Equation (10) can be expressed by using η, which can be used as a micro- and macro-scale correlation parameter to evaluate the influence of size effects.
(11)$$\sigma \left(\varepsilon ,\eta \right)=\eta m\tau_{R}\left(\varepsilon \right)+\left(1-\eta \right)\;\left(M\tau_{R}\left(\varepsilon \right)+\frac{k\left(\varepsilon \right)}{\sqrt{d}}\right)$$

The hybrid material model has two extremes, at size factor η = 0 and size factor η = 1, which is the flow stress that can be represented by the polycrystalline model and the monocrystalline model, respectively. From Equation (11), it can be seen that the hybrid-material model is represented as two components, the size dependent part and the size independent part. The hybrid material model can be represented as:(12)$$\{\begin{array}{c} \sigma \left(\varepsilon \right)=\sigma_{\mathrm{ind}}+\sigma_{\mathrm{dep}}\\ \sigma_{\mathrm{ind}}=M\tau_{R}\left(\varepsilon \right)+\frac{k\left(\varepsilon \right)}{\sqrt{d}}\\ \sigma_{\mathrm{dep}}=\eta \left(m\tau_{R}\left(\varepsilon \right)-M\tau_{R}\left(\varepsilon \right)-\frac{k\left(\varepsilon \right)}{\sqrt{d}}\right) \end{array}$$

As a result, the commonly used polycrystalline material model can be used to represent the hybrid material model, but the effect of size on flow stress needs to be subtracted.

The tensile results can be used to analyze the deformation behavior during the micro-wire deforming process and validate the hybrid material structure model proposed in this paper. The value of η varies from 0.10 to 0.46 depending on the specimen diameter as well as the grain size, respectively. According to the Hollomon formula, the pattern of reinforcement from the yield point to the cervical contraction can be expressed as follows: (13)$$\tau_{R}\left(\varepsilon \right)=K\varepsilon^{n}$$
(14)$$k\left(\varepsilon \right)=a\varepsilon^{b}$$

In Equation (13), n is work-hardening rate, K is strength factor, ε is true strain. The magnitude of the work-hardening rate n, which indicates the strain-reinforcing capacity of a metallic material and can be used to predict the resistance to further plastic deformation. The effect of polycrystalline grain boundaries on the flow stress can be expressed by the empirical formula, Equation (14), the a and b values for polycrystals are given (a_poly_ = 448.3/MPa, b_poly_ = 0.443) [39], and the value for single-crystals are listed in Table 2.

The flow stress curve can be determined using the exponential rule. We use the least square method to calculate the uncertainty coefficient in this hybrid material structure model. The resulting critical analysis flow stress is expressed as Equation (15):(15)$$\{\begin{array}{c} m\tau_{R}=788.4\varepsilon^{0.541}\\ \sigma =78.84\times \varepsilon^{0.541}+403.5\times \varepsilon^{0.443}\;\;\;\left(d=5.4\;\mu m,\eta =0.1\right) \end{array}$$

Figure 7 shows the calculation results of the hybrid material model, which includes size-related items and size-independent items. Specimens with different grain sizes and wire diameters can be characterized by calculating their size factor to characterize the flow stress profile; increasing the size factor contributes to the debilitating effect of the size dependent part. The simulation results show that the deformation model changes from a single-crystal model to a polycrystal model as η changes from 0 to 1. Moreover, the simulation results of the hybrid material model calculations presented in this paper are in good agreement with the experimental results, indicating that this hybrid material model is feasible in simulating flow stress during the tensile deformation of the material. Because of the limitations of the grain growth kinetics, the interval in which the grain size can be tuned is limited, and the K and n values obtained by experimental means are quite different from the single-crystal conditions at larger specimen level. The model proposed in this paper is well matched for samples below 200 μm.

## 5. Conclusions

The size effect on mechanical properties and deformation behavior of pure copper wires are investigated. The results show that a decrease in wire diameter leads to a reduction in tensile strength, and this change is pronounced for large grains. The elongation of the material is in linear correlation to size factor D/d, at the same wire diameter, more grains in the section bring better plasticity. A surface model combined with the theory of single crystal and polycrystal is established, based on the relationship between specimen/grain size and tensile property. The simulated results show that the flow stress at the micro-scale is in the middle of the single crystal model (lower critical value) and the polycrystalline model (upper critical value). Moreover, the simulation results of the hybrid model calculations presented in this paper are in good agreement with the experimental results under 200 μm. 

## Figures and Tables

**Figure 1 materials-13-04563-f001:**
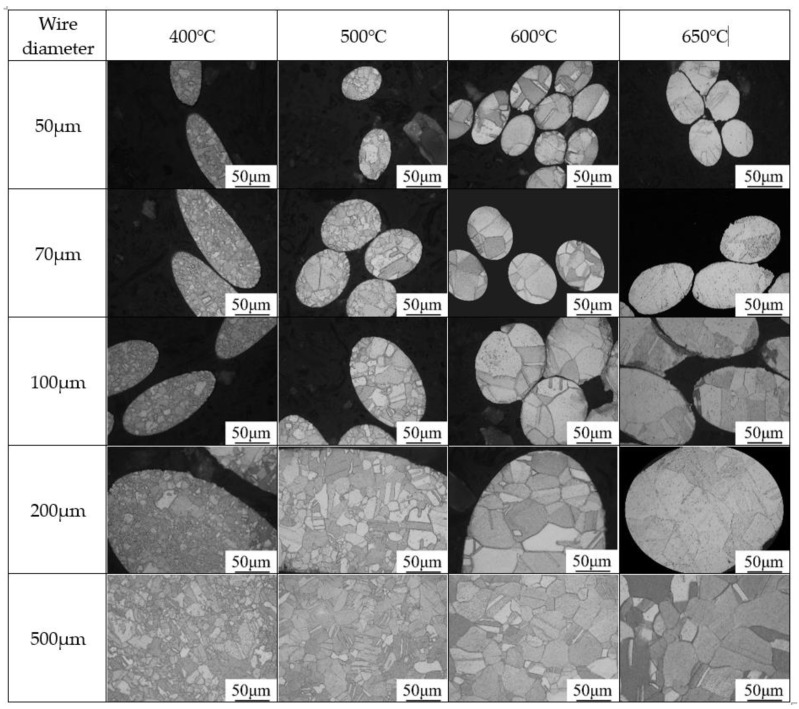
Optical microscopy of specimen with different grain sizes in cross section.

**Figure 2 materials-13-04563-f002:**
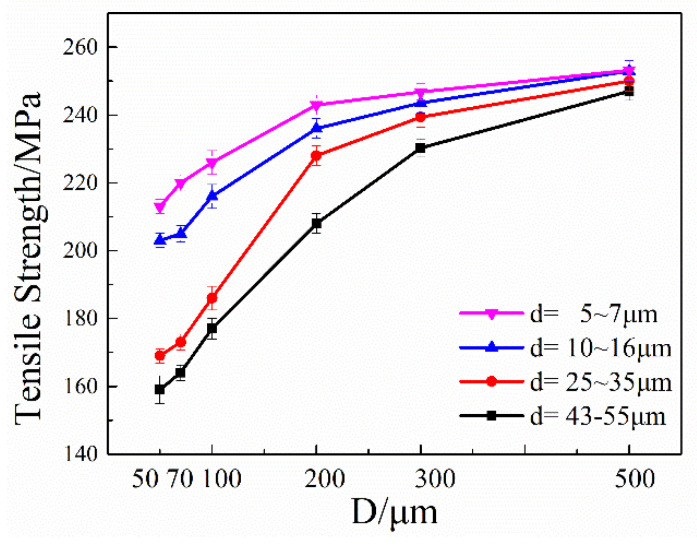
Tensile strength of specimens with different specimen and grain sizes (wire diameter: 50 ~ 500 μm, grain size: 6.5 ~ 50 μm).

**Figure 3 materials-13-04563-f003:**
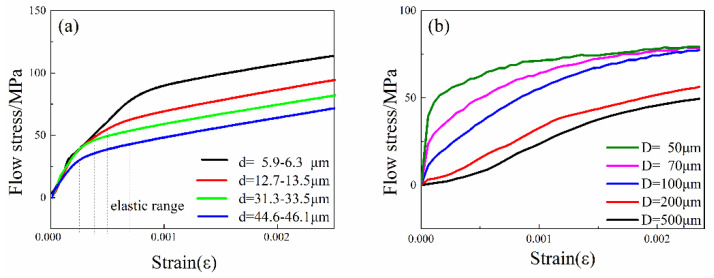
Effect of grain and specimen size on flow stress. (**a**) Flow stress curves of different grain size specimens with the same diameter (200 μm); (**b**) Flow stress curves of different wire diameter specimens with the same grain size (5–6.3 μm).

**Figure 4 materials-13-04563-f004:**
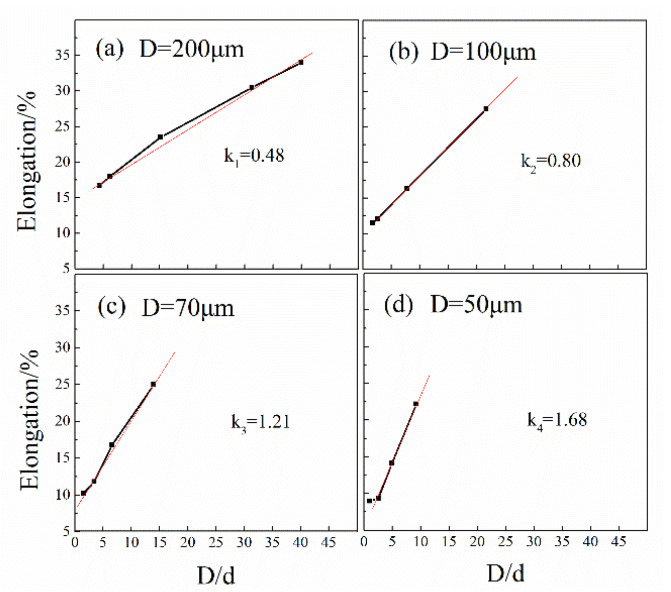
Tensile elongation of specimens with different size factors.

**Figure 5 materials-13-04563-f005:**
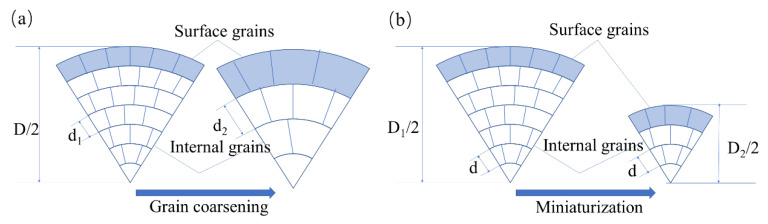
Grain size effects and specimen size effects with the decreasing of the scale. (**a**) Grain size effects and (**b**) specimen size effects.

**Figure 6 materials-13-04563-f006:**
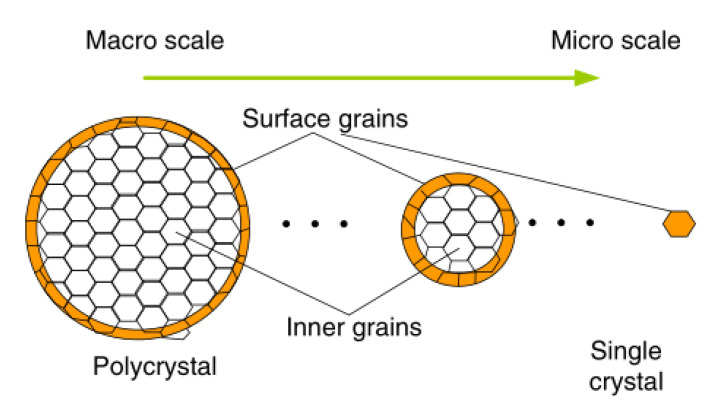
Particle distribution in the material cross-section varies with decreasing specimen size.

**Figure 7 materials-13-04563-f007:**
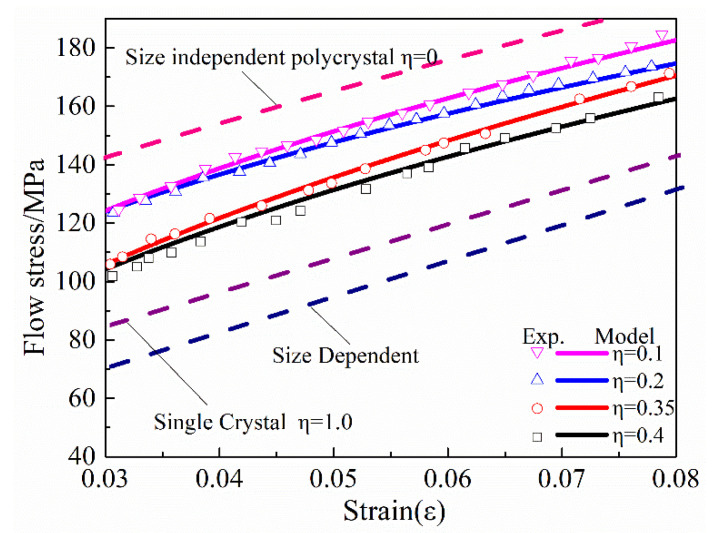
Comparison of calculated and experimental results for 200 μm diameter fine wires. The figure includes a size-related and a size-independent section.

**Table 1 materials-13-04563-t001:** Average grain size after annealing.

Annealing Temperature/°C	Wire Diameter, D/μm					
	50 μm	70 μm	100 μm	200 μm	300 μm	500 μm
400	5.4 ± 1.9	5.0 ± 1.4	4.6 ± 1.3	6.4 ± 2.1	7.1 ± 2.2	7.3 ± 2.4
500	10.1 ± 2.7	10.6 ± 2.9	12.9± 3.1	13.2 ± 3.3	15.8 ± 3.3	16.7 ± 3.9
600	19.7 ± 3.7	20.4 ± 3.2	39.1 ± 5.5	32.4 ± 3.7	35.1 ± 4.5	33.7 ± 4.0
650	43.1 ± 6.8	46.4 ± 7.5	50.8 ± 7.2	46.1 ± 6.9	52.3 ± 7.3	55.0 ± 7.2

**Table 2 materials-13-04563-t002:** Values of k and n obtained through testing.

D/μm	50	70	100	200	300	500
K/MPa	725.6	757.5	765.1	788.4	753.7	707.0
n	0.506	0.532	0.541	0.534	0.499	0.500

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
