# Peer review of "Size Effect on Mechanical Properties and Deformation Behavior of Pure Copper Wires Considering Free Surface Grains"

_materials, 2020, doi:10.3390/ma13204563_

Round 1

Reviewer 1 Report

The authors provide a paper focusing on the size effect on mechanical properties and deformation behavior of pure copper wires. The paper presents some significant results. Nevertheless, I am skeptical about some testing and the way to commenting the results. For these reasons, I recommend MAJOR revisions.

  • In the introduction, I would expand with the size effects involving different class of materials including crystalline and amorphous metals see for instance doi.org/10.1016/j.actamat.2011.11.023. These papers have to be included also when commenting the results.
  • Following with the previous point, I would like that authors comment on the critical size to activate size effects in relation with the materials type. As a matter of facts, most of the materials exhibit size effect for thickness below 500 nm as in the previous Refs. The authors must carefully comment on that.
  • The authors must comment on the possibility to have some compositional changes reducing the size of the sample. EDX experiments are requested. Moreover, the authors can comment on the present paper https://doi.org/10.1038/s41598-019-49910-7 reporting compositional change for thin ZrNi metallic glasses.
  • The authors did not investigate the effect of grain boundaries and interfaces reducing the diameter of the wire? This effect must be decoupled with the size effects.
  • The authors must comment on the evolution of elastic modulus as a function of the size of the wire. It will be a good check to see if the experiments are performed correctly.

Reviewer 2 Report

The manuscript by Hou et al is devoted to grain size/wire diameter ratio effect on tensile strength, those are studied both theoretically and experimentally.

I find the research interesting and comprehensive. The suggested model seem to be successful in adequate description of the discussed phenomena. However the experimental part is not adequately described. The presentation is also far from ideal. Please address my comments below:

1.Experiment. It is not described what is meant by "grain size" and how was it determined from the photos of the cross-sections. Looking e.g. at Fig. 1 (right-bottom frame) I see very huge variations of size from grain to grain. However in the right-bottom cell of Table 1 it is written that the grain size is 55+/-7 micrometers. I can not believe in it. Please provide the algorithm., how these figures were obtained.

2. Experiment. The authors do indicate purity 99.7% but do not indicate what is the impurity and who is a supplied. This makes the results potentially irreproducible.

3. Experiment. How was the cross-section prepared and the microscopy image obtained?

4. Presentation. References. In page 1 it is written "experiment of Kals[7,15,20] ", however Refs 7 and 20 are not by Kals. The numbers of references are not ordered. Ref.5 - surnames are absent. Refs. 31 and 39 - why are the surnames capitalized?

5. English. The language quality has to be improved. I suggest the authors to use a special service.

6. Logic: once the model is postulated,  its limitations have to be shown. In this paper the authors only demonstrate the agreement between the model and experimental data for 200 micrometer thick wire. It does not prove anything and leaves a question "how general is the model?" open.

7. How many adjustable parameters are used?

To conclude. In the present the paper is not ready for publication in MDPI materials.

Round 2

Reviewer 1 Report

-

Reviewer 2 Report

In the revised manuscript the authors respond my queries.